# Why Does the Antioxidant Complex Twendee X^®^ Prevent Dementia?

**DOI:** 10.3390/ijms241613018

**Published:** 2023-08-21

**Authors:** Fukka You, Yoshiaki Harakawa, Toshikazu Yoshikawa, Haruhiko Inufusa

**Affiliations:** 1Division of Anti-Oxidant Research, Life Science Research Center, Gifu University, Yanagito 1-1, Gifu 501-1194, Japan; y@antioxidantres.jp (F.Y.); harakawa@antioxidantres.jp (Y.H.); 2Anti-Oxidant Research Laboratory, Louis Pasteur Center for Medical Research, Tanakamonzen-cho 103-5, Sakyo-ku, Kyoto 606-8225, Japan; 3Louis Pasteur Center for Medical Research, Tanakamonzen-cho 103-5, Sakyo-ku, Kyoto 606-8225, Japan; toshi@yoshikawalab.jp; 4School of Medicine, Kyoto Prefectural University of Medicine, Kajii-cho, Kawaramachi-Hirokoji, Kamigyo-ku, Kyoto 602-8566, Japan

**Keywords:** Alzheimer’s disease, oxidative stress, Twendee X^®^, antioxidant, dementia, mitochondria, reactive oxygen species

## Abstract

Alzheimer’s disease (AD) is a complex neurodegenerative disease characterized by cognitive and short-term memory impairments. The disease involves multiple pathological factors such as amyloid plaque formation, mitochondrial dysfunction, and telomere shortening; however, oxidative stress and diabetes mellitus are significant risk factors. The onset of AD begins approximately 20 years before clinical symptoms manifest; therefore, treating AD after symptoms become evident is possibly too late to have a significant effect. As such, preventing AD or using an effective treatment at an early stage is important. Twendee X^®^ (TwX) is an antioxidant formulation consisting of eight ingredients. TwX has been proven to prevent the progression to dementia in patients with mild cognitive impairment (MCI) in a multicenter, randomized, double-blind, placebo-controlled, prospective intervention trial. As well, positive data has already been obtained in several studies using AD model mice. Since both diabetes and aging are risk factors for AD, we examined the mechanisms behind the effects of TwX on AD using the spontaneous hyperglycemia model and the senescence model of aged C57BL/6 mice in this study. TwX was administered daily, and its effects on diabetes, autophagy in the brain, neurogenesis, and telomere length were examined. We observed that TwX protected the mitochondria from oxidative stress better than a single antioxidant. TwX not only lowered blood glucose levels but also suppressed brain neurogenesis and autophagy. Telomeres in TWX-treated mice were significantly longer than those in non-treated mice. There are many factors that can be implicated in the development and progression of dementia; however, multiple studies on TwX suggest that it may offer protection against dementia, not only through the effects of its antioxidants but also by targeting multiple mechanisms involved in its development and progression, such as diabetes, brain neurogenesis, telomere deficiency, and energy production.

## 1. Introduction

Age-related accumulation of oxidized products elevates the risk for humans to develop many cognitive diseases, including dementia. Alzheimer’s disease (AD) causes memory and cognitive dysfunctions that gradually worsen as the disease progresses. Since these conditions are irreversible, AD patients eventually lose the ability to live independently and require nursing care. In AD, the greatest risk factor is advanced age, followed by genetics, nutrition, physical activity, and environmental factors [1]. As the world’s population ages, dementia is expected to increase to an estimated 135 million patients by 2050. This number will increase further if people with possible dementia risk factors (such as obesity in middle age and diabetes) are included.

Despite these alarming statistics, there are currently no effective disease-modifying agents to treat age-related cognitive dysfunction and dementia. Since pharmaceuticals can only be administered once dementia has been diagnosed, their effectiveness may be quite limited after the onset of dementia. As such, until more effective treatments are developed, the key to reducing the incidence and effects of dementia is to proactively prevent its onset in the first place. AD is a progressive neurodegenerative disease characterized by cognitive and short-term memory impairment [1,2]. The etiology of AD is complex and includes multiple factors such as the formation of amyloid-β (Aβ) and phosphorylated tau (p-tau), neuroinflammation, synaptic dysfunction, mitochondrial dysfunction, telomere shortening, and oxidative stress [2,3,4]. Oxidative stress, in particular, is the most implicated, with many studies labeling AD as an oxidative stress disease [5,6,7,8,9]. There are several reasons for this; mitochondrial dysfunction, for example, is said to be an early signal of the onset of AD [2,10], and it is closely related to oxidative stress, including decreased ATP production and increased free radicals. Diabetes mellitus, a condition that has been branded as an oxidative stress disease, is also a significant risk factor for AD. In addition to oxidative stress, diabetes mellitus exhibits inflammation, mitochondrial dysfunction, and several similar neurodegenerative mechanisms [11]. In fact, epidemiological studies have found that 70–80% of patients with AD are diabetic or have abnormal blood glucose or insulin levels [12,13]. This strongly suggests that preventing diabetes is likely to reduce the incidence of AD.

If over-the-counter dietary supplement formulations consisting of multiple antioxidants can offer protection against dementia, then the onset of dementia can be prevented and its further progression can possibly be delayed. There is research being performed in this area, and although several studies have shown promising effects of antioxidants on dementia, no study so far has been successful in significantly reducing the risks of dementia using antioxidants [14].

Although it is a dietary supplement, Twendee X^®^ (TwX) has passed the same safety standards required of pharmaceuticals, including chromosomal aberration, toxicity, and mutation tests. TwX consists of eight active ingredients, and it is being investigated as a potential therapeutic option for cognitive and other diseases resulting from oxidative stress. The results of at least four studies are quite promising. TwX was shown to prevent mild cognitive impairment (MCI) in a multicenter, randomized, double-blind, placebo-controlled, prospective intervention trial consisting of Japanese participants [15]. In the second study, the effects of TwX were examined in a mouse model of AD with chronic cerebral hypoperfusion (CCH + APP23 mice), with observed loss of hippocampal neurons and increased expression of oxidative stress markers, p-tau, and phospho-α-synuclein. TwX improved motor coordination and working memory and restored the loss of hippocampal neurons. It also significantly improved cognitive impairment, reduced Aβ pathology and neuronal loss, and alleviated neuroinflammation and oxidative stress [16,17].

Additionally, TwX significantly inhibited Aβ Oligomer(AβO)-induced phosphorylation of tau protein, improved cell morphology, suppressed intracellular reactive oxygen species (ROS), and improved cell viability in the SH-SY5Y cell line [18]. Finally, in a mouse model of ischemic stroke, TwX reduced infarct size, showing neuroprotective effects [19]. In order to further understand the reasons for these results, this study examines the effects of TwX on AD onset and progression using a spontaneous hyperglycemia model in aged mice. We believe that using mice that have developed hyperglycemia in as natural a state as possible without the use of genetic modification is likely to produce results that are unbiased and truly demonstrate the preventive effect of TwX.

## 2. Results and Discussion

### 2.1. Antioxidant Capacity of TwX

Vitamin C (VC) is one of the eight active antioxidants in TwX [20,21]. The antioxidant capacity of TwX (60 mg/mL, n = 6) and VC alone (20.5 mg/mL, n = 6) in the same concentration as is present in TwX was measured by the OXY adsorbent test. Our analysis found that a 60 mg/mL dose of TwX had 5.1 times higher antioxidant capacity than 20.5 mg/mL of VC alone (Figure 1).

It is known that exposure of cells to the oxidant hydrogen peroxide (H_2_O_2_) causes the production of reactive oxygen species (ROS). Human hepatocellular carcinoma cells, HepG2 cells, were treated with H_2_O_2_ (100 μM), and changes in redox status were examined as relative absorbance values compared to the non-treated group. Exposing HepG2 to 100 µM H_2_O_2_ stimulated an increase in both mitochondrial ROS (mtROS) by 69% and cellular ROS (cROS) by 68% compared to normal cells. As well, manganese superoxide dismutase (Mn-SOD) and copper/zinc superoxide dismutase (Cu/Zn-SOD) decreased by 32% and 31%, respectively, and this was accompanied by an increase in the GSSG/GSH ratio. TwX reduced both types of H_2_O_2_-induced ROS as well as the GSSG/GSH ratio and increased the antioxidant levels in cells and mitochondria. TwX in concentrations of 60, 120, and 240 µg/mL resulted in an increase in ROS scavenging and SOD activity in cells, with the optimal effect observed at 60 µg/mL (Table 1).

ROS is the general term for radical or non-radical oxygen derivatives [22], including free radicals such as superoxide (O_2_^•−^), hydroxyl radical (OH^•^), and peroxides (hydrogen peroxide/H_2_O_2_). ROS are highly reactive and can readily lose electrons (oxidation) in the presence of cellular biomolecules, setting off a chain reaction that ultimately damages the cellular structure. Among ROS, OH^•^ has been reported to directly attack the DNA backbone and cause DNA damage [23,24]. Mitochondria are the primary site of ROS production. Many subunit syntheses in the mitochondrial respiratory chain complex (except complex II) are based on mitochondrial genes (mtDNA), and ROS can greatly affect the transcription of mtDNA-encoded proteins and RNAs. Thus, mitochondria with oxidized mtDNA dysfunction produce large amounts of ROS, which further impair mitochondrial function, thus establishing a continuous cycle of oxidation. This chronic oxidative stress eventually leads to severe nuclear DNA damage and cell death [25].

ROS are also closely associated with inflammation, which is the host’s protective immune response against foreign pathogens and promotes tissue repair. The increase in ROS production by polymorphonuclear neutrophils (PMNs) at the site of inflammation is primarily due to tissue damage and endothelial dysfunction. Inflammation, in turn, promotes the production of ROS, and the lack of an efficient ROS removal system can accelerate this inflammatory cycle. Furthermore, there is a reciprocal relationship where ROS induces the generation of an inflammatory response [26,27], resulting in a synergistic effect.

Previous reports have identified oxidative damage as a significant factor in the pathogenesis and progression of AD. Recent studies have also highlighted the role of oxidative stress in the prodromal stages of AD, including MCI [28]. Aging, the greatest risk factor for AD, is the result of accumulated oxidative stress. Aβ accumulated in the brain after the onset of AD also generates free radicals and causes inflammation in the surrounding neurons, leading to mitochondrial dysfunction and eventual neurodegeneration. Thus, AD is the result of a persistent cycle of oxidative stress, inflammation, and mitochondrial dysfunction.

Given the significant contribution of oxidative stress to the etiology and progression of AD, researchers have conducted clinical trials to investigate the potential effects of antioxidants. Vitamin E (VE), a fat-soluble vitamin with antioxidant properties, may interrupt cell-damaging reactions by interacting with cell membranes to trap free radicals. Clinical trials using high doses of VE (2000 IU/day) reported that it slowed disease progression in patients with moderate AD more than in patients who received a placebo [29,30]; however, other subsequent studies using VE have failed to replicate these outcomes [14]. Ascorbic acid (AsA), another antioxidant, has been found to inhibit oxidative stress and excessive secretion of inflammatory factors, thereby extending lifespan. Furthermore, in a mouse model, AsA has been found to positively modulate both inflammatory and immune aging, which are two features of biological aging; however, to date, no results in humans have been obtained [31].

In the present study, a combined formulation of TwX demonstrated higher antioxidant capacity compared to an equivalent dose of VC. It also suppressed intracellular ROS and increased the activity of SOD, an antioxidant enzyme. Glutathione normally exists in the reduced form (GSH) in vivo but is converted to the oxidized form (GSSG) by stimuli such as oxidative stress. TwX reduced the ratio of oxidized glutathione (GSSG) to reduced glutathione (GSH), indicating its ability to protect cells and mitochondria from oxidation. Additional evidence of TwX’s effectiveness was demonstrated through its ability to scavenge free radicals (CH_3_^•^), (OH^•^), and (O_2_^•−^) generated from the H_2_O_2_/NaOH/DMSO reaction system using ESR [32]. In particular, (OH^•^) and (O_2_^•−^) scavenging can be eliminated at low doses. Unlike other formulations containing high concentrations of AsA, TwX did not produce adverse effects such as increasing the signal height of (CH_3_^•^) [33,34]. These results may be attributed to the novel blend of eight active ingredients in TwX, which effectively inhibits the formation of ascorbyl radicals. In other studies, TwX significantly improved cognitive dysfunction and impaired coordination in vitamin E-deficient mice [35].

Considering previous antioxidants have been single agents and have limited effects, TwX, which contains multiple agents, is a novel antioxidant.

### 2.2. Mitochondrial Energy Production

Metabolomic analysis was performed on HepG2 cells, a human hepatocellular carcinoma cell line, to investigate the effects of TwX stimulation. Unstimulated cells served as the control. Cells were stimulated with TwX (60 µg/mL) for 1 h. Capillary electrophoresis–time-of-flight mass spectrometry (CE–TOFMS) revealed 233 peaks (111 cations and 122 anions). Liquid chromatography (LC–TOFMS) detected 93 peaks (58 positive and 35 negative). The relative area ratio between each peak was calculated, and Welch’s *t*-test was performed to determine significant differences. Next, hierarchical cluster analysis (HCA) was performed using the metabolome data, resulting in a clustered heat map displaying 326 biochemicals in HepG2 lysates (Figure 2).

Among the substances detected, metabolite levels related to the glycolytic and tricarboxylic acid cycles (TCA) are shown (Appendix A) and depicted in the metabolic pathway diagram in Figure 3. Notably, the concentrations of adenosine triphosphate (ATP) and adenosine diphosphate (ADP) increased in the TwX-treated group compared to the control group.

Mitochondria are essential intracellular organelles responsible for most of the ATP production needed for energy-requiring processes and cellular pathways. Mitochondria also produce coenzymes, which are important for the neurotransmitter acetylcholine and the conversion of fatty acids into energy sources. On the other hand, mitochondria also produce several byproducts such as ROS superoxide (O_2_^•−^), hydroxyl radicals (OH^•^), and hydrogen peroxide (H_2_O_2_).

The brain, which utilizes about 25% of the body’s total glucose [36], relies on glucose for energy conversion through glycolysis and mitochondrial oxidative phosphorylation to support synaptic transmission. However, aging decreases glucose uptake from neurons and reduces the activity of the electron transport system. Furthermore, most of the neuronal energy transfer pathways occur in the mitochondria, and elevated oxidation interferes with this energy conversion. These mitochondrial abnormalities are a hallmark of brain aging and neurodegenerative diseases [37] and are particularly prominent in AD, where they have been observed along with impaired mitochondrial dynamics (increased mitosis and decreased fusion), altered mitochondrial morphology and mitochondrial gene expression, increased free radical production and lipid peroxidation, and decreased cytochrome c oxidase (COX) and ATP production. These mitochondrial abnormalities lead to the progression and pathogenesis of AD [4,38,39,40,41]. In fact, postmortem studies of AD tissues have revealed dysfunction of the TCA cycle, impaired electron transfer and oxidative phosphorylation, and changes in mitochondrial morphology, suggesting disruption of mitochondrial function [42,43]. These dysfunctions are associated with changes in redox status due to free radicals and H_2_O_2_. Both bioenergetic deficits and chronic oxidative stress significantly contribute to brain aging and cognitive decline associated with AD. TwX was suggested to protect mitochondria from redox state changes caused by free radicals and H_2_O_2_, thereby preventing mitochondrial dysfunction and stimulating energy production. By targeting these mechanisms, TWX may offer potential benefits in mitigating the progression and pathogenesis of AD.

### 2.3. Effect on Diabetes

Given that diabetes is a significant risk factor for AD, the effect of TwX on diabetes was examined. Previous studies have used TwX concentrations of 20 mg/kg/day (AD and stroke model mice [16,17,19]) and 40 mg/kg/day (cognitive function experiments in vitamin E-deficient model mice [35]). However, in this study, 40 mg/kg/day was used since the mice were old and had diabetic disease.

The blood oxidative stress level of 35-week-old spontaneously hyperglycemic mice was significantly higher than that of normal mice of the same age (NC). The TwX group had lower oxidative stress levels than the TwX non-treated group (Control), although the difference was not significant (Figure 4).

At 41 weeks old, mice were fasted for 16 h, and a glucose tolerance test was performed to verify changes in blood glucose levels. The blood glucose level of the non-treated group (Control) was significantly higher than that of normal mice of the same age (NC), while the blood glucose level of the TwX-treated (40 mg/kg/day) group did not show rapid changes and was lower than that of the Control group (Figure 5). These findings suggest that TwX can potentially improve hyperglycemia, which would be beneficial for diabetes, which is a known risk factor for dementia.

Diabetes, a treatable metabolic disease, has now been shown to increase the risk of AD, vascular dementia, and all other types of dementia [44,45]. Diabetes, as with AD, progresses and develops gradually over a long period of time and causes mild to moderate cognitive impairment not attributable to any other factor. Several common neurodegenerative mechanisms have been identified in diabetes, including oxidative stress, mitochondrial dysfunction, and inflammation. Although the brain is an immunologically endowed organ, there are reports of crosstalk between peripheral and central inflammation. Chronic hyperglycemia elevated inflammatory markers and contributed to increased ROS generation, leading to metabolic inflammation and the induction of mitochondrial oxidative stress, endoplasmic reticulum (ER) stress, and autophagy defects. In parallel, blocking autophagy may be associated with proinflammatory signaling through oxidative stress pathways and NF-κB-mediated inflammation [46]. TwX has been found to improve hyperglycemia and suppress the increase in ROS caused by persistent hyperglycemia, suggesting that it may inhibit vascular endothelial damage caused by diabetes and may prevent cognitive decline.

### 2.4. Effect on Autophagy

To examine the effect of TwX on autophagy, brains from 70-week-old spontaneously hyperglycemic mice were harvested, and the expression level of LC3 protein, an autophagy marker, was assessed using western blotting.

Compared to the TwX non-treated group (Control), there was a significant increase in the LC3-II/LC3-I ratio in the TwX-treated group, indicating enhanced autophagy (Figure 6).

Autophagy is a cellular process that helps maintain the balance of healthy cells, organelles, proteins, and nutrients in an organism. It can occur selectively or non-selectively, depending on whether the target is a specific cellular component or an entire cell [2]. However, the efficiency of autophagy decreases with age, and autophagy dysfunction has been reported in various neurodegenerative diseases. Autophagy dysfunction can occur at different stages of the autophagy mechanism and may contribute to the formation of intracellular aggregates and ultimately neuronal cell death [47].

Recent studies have highlighted the role of autophagy dysfunction, particularly in the context of Aβ- and p-tau-induced pathology in AD. With age-dependent increases in Aβ and p-tau, levels of several autophagy proteins (ATG, LC3-1,-2) and mitophagy proteins (PINK1, Parkin, P62, BNIP3, FUNDC1, LC3-1, OPTN, and TBK1) are decreased [2]. Maintaining normal autophagy is crucial since autophagy dysfunction, along with mitochondrial abnormalities and synaptic damage, is a major cellular event in Aβ accumulation and the onset, progression, and pathogenesis of AD. LC3 microtubule-associated proteins play an essential role in the autophagy pathway, and the absence of LC3 protein expression in the diabetic brain, a risk factor for AD, is significant. However, in the TwX-treated group, LC3 protein expression was observed, suggesting that TwX may help maintain autophagy function.

### 2.5. Effect on Telomere

Telomere length in tail DNA from 65-week-old spontaneously hyperglycemic mice was significantly longer in the TwX-treated group than in non-treated mice of the same age (Control) group (Figure 7).

Telomere length is considered a biological marker of age, and telomeres shorten as cells divide during the aging process [48]. Recent reports have suggested that telomere shortening is a potential factor in the development of aging-related diseases and AD [49,50,51] and that shorter telomeres tend to be present in dementia patients [52].

Telomeres are repetitive DNA sequences that have the primary function of protecting the ends of chromosomes from degrading and fusing with adjacent chromosomes. Telomeres also play an important role in genome replication, repair, and maintenance [53]. Many reports have associated telomere shortening with neurodegeneration and neurodegenerative disease processes [52,54]. It is implicated in cognitive dysfunction and amyloid pathology in AD and plays an important role in the pathogenesis of AD through mechanisms of oxidative stress and inflammation [55]. The elongation of telomere length with TwX is the result of improvements in various factors involved in AD development and progression, such as reduction of ROS and protection of mitochondria; whether telomere lengthening prevents AD is unknown. However, the results show that TwX results in telomere lengthening.

### 2.6. Neurogenesis

In the hippocampal dentate gyrus of 56-week-old mice, Tuj-1, a marker of neurons during early development, was markedly decreased. On the other hand, in the TwX-treated group of the same age, the age-related decrease in Tuj-1 in the hippocampal dentate gyrus was suppressed, and numbers similar to those of younger (6-week-old) mice were maintained (Figure 8).

Neurogenesis is well-known to occur throughout life in adult mammalian brain regions, including humans, and is determined by neurotrophins, growth factors, hormones, exercise, ischemia, and trauma. However, with aging, the capacity for neurogenesis declines, and its number decreases [56]. In the present study, we observed fewer newborn neurogenic cells due to aging. Not only aging but also neurotoxic degeneration causes a decrease in newborn neurogenic cells. Viable progenitor cells have been found to remain in aged normal and AD brains.

Progenitor cells from AD and aged normal control specimens can differentiate into tubulin-positive and Tuj-1-positive neurons and GFAP-positive astrocytes. However, there are significantly fewer viable progenitor cells in the hippocampus of AD patients compared to age-matched normal controls [57]. This translates into a lack of function in AD brains to compensate for neurons lost due to aging and neurodegeneration.

The results of the present study suggest that TwX suppresses the decrease of newborn neurogenic cells in the brains of aged mice, although whether TwX can suppress the decrease of newborn neurogenic cells in the AD brain is not known. However, TwX can certainly affect neurogenesis in the brain. Further studies are needed to examine the effects of TwX on progenitor cells and neurogenic cell fields in the AD brain, which may assist in preventing and treating AD.

## 3. Materials and Methods

### 3.1. Materials

TwX is a mixture consisting of L-glutamine (34.6 wt%), ascorbic acid (34.2 wt%), L-cystine (18.2 wt%), coenzyme Q10 (3.6 wt%; AQUA Q10 P40-NF, Nissin Pharmaceutical, Tokyo, Japan), succinic acid (3.6 wt%), fumaric acid (3.6 wt%), riboflavin (1.5 wt%; Bislase inj; Toa Eiyo, Tokyo, Japan), and niacin amid (0.7 wt%). The TwX mixture was dissolved with MiliQ water (Sigma-Aldrich, Tokyo, Japan) and stored at 4 °C until use.

### 3.2. Antioxidant Measurement of Solutions

The antioxidant capacity of TwX and vitamin C solutions was determined using the OXY adsorption test (Diacron International Srl, Grosseto, Italy). The OXY adsorption test measures the ability of a sample to scavenge hypochlorous acid (HClO), one of the most potent ROS produced by white blood cells.

TwX solution (60 mg/mL) and the same amount of Vitamin C solution (20.5 mg/mL) contained in TwX were prepared on the day of the assay, and the OXY adsorption test was performed according to the kit’s instructions. Both tests were performed in duplicate, and the average value was used.

### 3.3. Intracellular and Intramitochondrial Redox Changes In Vitro

To characterize the antioxidant properties of TwX, parameters of the redox state following the induction of oxidative stress by hydrogen peroxide (H_2_O_2_) in HepG2 cells were assessed. This study was conducted by ICDD-sas (France) using the Mitosafe^®^ technology, developed by ICDD-sas, which consists of the study of integrated mechanisms controlling mitochondrial function within live cells. Analyses were carried out with TwX, measuring its antioxidant properties toward parameters of the redox state following the induction of oxidative stress by H_2_O_2_ (100 µM) in HepG2 cells. TwX was compared to the reference antioxidant agent, Trolox. The following outcomes were measured: the mitochondrial and cellular reactive oxygen species (mtROS, cROS) production rate, Mn-SOD and Cu/Zn-SOD activities, and the GSSG/GSH ratio. All data were recorded by the SkanIt software (Thermo), and the Student’s *t*-test was used for statistical analysis.

### 3.4. Metabolome Analysis

#### 3.4.1. Cell Preparation

HepG2 cells were cultured in 12 dishes (Φ10 cm) in Dulbecco’s Modified Eagle medium (DMEM, Fujifilm Wako, Osaka, Japan) with 10% FBS (Global Life Sciences Solutions, Marlborough, MA, USA) in a CO_2_ incubator at 37 °C. After reaching confluency, TwX (60 µg/mL) was added to HepG2 cells in three dishes. After one hour, each TwX-stimulated and non-stimulated (Control) dish served as one sample and was subjected to metabolite extraction for CE–TOFMS (n = 3) and LC–TOFMS (n = 3) analysis.

#### 3.4.2. Metabolite Extraction for CE–TOFMS

The culture medium was aspirated from the dish, and the cells were washed twice with a 5% mannitol solution (10 mL and 2 mL for the first and second wash, respectively). The cells were then treated with 800 µL of methanol and incubated at room temperature for 30 sec to suppress enzyme activity. Next, 550 µL of Milli-Q water containing internal standards (H3304-1002, Human Metabolome Technologies, Inc. (HMT), Tsuruoka, Yamagata, Japan) was added to the cell extract, followed by further incubation at room temperature for 30 s. The cell extract was then centrifuged at 2300× *g* at 4 °C for 5 min, after which 700 µL of the supernatant was centrifugally filtered through a Millipore 5-kDa cutoff filter (UltrafreeMC-PLHCC, HMT) at 9100× *g* at 4 °C for 120 min to remove macromolecules. Subsequently, the filtrate was evaporated to dryness under a vacuum and reconstituted in 50 µL of Milli-Q water for metabolome analysis at HMT.

#### 3.4.3. Metabolite Extraction for LC–TOFMS

The culture medium was aspirated from the dish, and the cells were washed twice with a 5% mannitol solution (10 mL and 2 mL for the first and second wash, respectively). The cells were then treated with 1300 µL of ethanol containing internal standards (H3304-1002, Human Metabolome Technologies, Inc. (HMT), Tsuruoka, Yamagata, Japan), transferred into a tube, and re-suspended in 1000 µL of Milli-Q water. The cell solution was sonicated for 5 min and centrifuged at 2300× *g* at 4 °C for 5 min. Subsequently, the supernatant was evaporated to dryness under nitrogen and reconstituted in 100 µL of 50% isopropanol (*v*/*v*) for metabolome analysis at HMT.

#### 3.4.4. Analysis

Metabolome analysis was conducted according to HMT’s Dual Scan package, using CE–TOFMS and LC–TOFMS based on the methods described previously [58,59]. Briefly, CE–TOFMS analysis was carried out using an Agilent CE capillary electrophoresis system equipped with an Agilent 6210 time-of-flight mass spectrometer (Agilent Technologies, Inc., Santa Clara, CA, USA). LC–TOFMS analysis was carried out by an Agilent 1200 HPLC pump with an Agilent 6210 time-of-flight mass spectrometer (Agilent Technologies). The systems were controlled by Agilent G2201AA ChemStation software version B.03.01 for CE (Agilent Technologies) and MassHunter for LC (Agilent Technologies). The spectrometer was scanned from *m*/*z* 50 to 1000, and peaks were extracted using MasterHands automatic integration software version 2.17.1.11 (Keio University, Tsuruoka, Yamagata, Japan) in order to obtain peak information including *m*/*z*, peak area, migration time (MT) for CE–TOFMS, and retention time (RT) for LC–TOFMS analyses [60]. Signal peaks corresponding to isotopomers, adduct ions, and other product ions of known metabolites were excluded, and the remaining peaks were annotated according to HMT’s metabolite database based on their *m*/*z* values and MTs or RTs. Areas of the annotated peaks were then normalized to internal standards and sample amounts in order to obtain relative levels of each metabolite. HCA and principal component analysis (PCA) [61] were performed by HMT’s proprietary MATLAB and R programs, respectively. Detected metabolites were plotted on metabolic pathway maps using VANTED software [62].

### 3.5. Animals

All experimental procedures were approved by the Animal Committee of the Gifu University Graduate School of Medicine. In all experiments, C57BL/6JJcl male mice obtained from Japan SLC (Shizuoka, Japan) were used throughout this study. Mice were maintained in a temperature- and humidity-regulated room (23 ± 3 °C, 50 ± 10%) on a 12-h light–dark cycle and allowed free access to food and water unless otherwise mentioned.

### 3.6. Preparation of Old Spontaneous Hyperglycemia Model Mice

C57BL/6J mice (5 weeks old, male) were acclimatized for 7 days and then fed a high-fat diet HFD-60 (Oriental Yeast Co., Ltd., Tokyo, Japan) ad libitum, except for the negative control (NC group, n = 2). Once the fasting blood glucose level of the mice reached 100 mg/dl, the mice were divided into two groups: a non-TwX group (sterile water, Control group, n = 12) and a TwX group (40 mg/kg/day, TwX group, n = 12) so that the fasting blood glucose level and body weight of the mice were equalized. After grouping, each mouse was given the same volume of TwX-containing solution or sterile water by forced oral gavage once a day until it reached the age required for the experiment. These mice were labeled as old spontaneously hyperglycemic mice.

### 3.7. Glucose Tolerance Test

Among the 41-week-old spontaneously hyperglycemic mice raised, those within ±5.0 g of the mean body weight of each group were subjected to the following tests: Control group, n = 10; TwX group, n = 8; and NC group, n = 2. After each mouse was fasted for 16 h, blood samples were collected. Then, in the glucose tolerance test, all mice were given 1 g/kg glucose solution (Fuso Pharmaceutical Industries, Ltd., Osaka, Japan) intraperitoneally, and blood glucose levels were measured at scheduled intervals (15, 30, 60, 120, and 180 min after glucose loading) with a self-test glucose meter (Nipro Corporation, Osaka, Japan).

### 3.8. Measurement of Oxidative Stress in the Blood

The d-ROMs test (Diacron-Reactive Oxygen Metabolites) was used to measure oxidative stress in mouse blood.

The d-ROMs test quantifies in vivo oxidative stress by capturing hydroperoxide (ROOH), a metabolite produced when ROS and free radicals oxidize body components. The old spontaneously hyperglycemic mice within ±5.0 g of the mean body weight of each group (Control group, n = 10; TwX group, n = 8; NC group, n = 2) were fasted for 16 h, blood was collected, and plasma was measured according to the d-ROMs kit instructions.

### 3.9. Western Blotting

The 70-week-old spontaneously hyperglycemic mice were euthanized by cervical dislocation, and the hippocampal dentate gyrus was removed. Samples were homogenized and lysed, and total proteins were extracted for Western blotting. Rabbit anti-LC3 (catalog no. PM036, Medical & Biological Laboratories; ×1000 dilution with 1% skimmed milk) and rabbit anti-IgG pAb-HRP (MBL, code no. 458; ×5000 dilution with 1% skimmed milk) were used as primary and secondary antibodies, respectively. Western blotting was performed by Genostaff Co., Ltd. (Tokyo, Japan) according to the following procedure: lysates (50–100 μg protein) were fractionated by SDS–polyacrylamide gel electrophoresis (PAGE) and then transferred onto polyvinylidene fluoride sheets (PVDF) membranes. The transferred PVDF was blocked with 10% skimmed milk dissolved in PBS overnight at 4 °C, then incubated for 1 h at room temperature with the primary antibodies. The membrane was incubated with a secondary antibody for 1 h at room temperature. After removing the extra reagent from the membrane, it was exposed to an X-ray film in a dark room for 3 min, and bands were detected with the C-DiGit Blot Scanner (LI-COR, Lincoln, NE, USA). Band intensity was evaluated using ImageJ version 1.53k (National Institutes of Health, Bethesda, MD, USA) as previously described [63].

### 3.10. Measurement of Relative Telomere Length

The tails of the 65-week-old spontaneously hyperglycemic mice, which were fed ad libitum on an HFD-60 diet, were cut to a length of 5 mm at the tip and immediately quenched with liquid nitrogen. The genome extraction procedures were outsourced to BioGate Co., Ltd. (Gifu, Japan). The MagExtractor Genome (TOYOBO NPK-1, Osaka, Japan) kit was used to extract the genome from the tails according to the manufacturer’s instructions. The frozen mouse tails were filled with 850 µL of lysate and homogenized using a micro-homogenizer. The samples were centrifuged at 10,000 rpm for 5 min, and 40 µL of magnetic beads were added to the supernatant and mixed vigorously for 10 min. The supernatant was removed with a magnetic stand, washed twice with 900 µL of washing solution and twice with 70% ethanol, and the DNA was separated and eluted with 60 µL of sterile water. The telomere length from the isolated DNA was measured with the Relative Mouse Telomere Length Quantification qPCR Assay Kit (ScienCell M8908, Carlsbad, CA, USA) and KOD-SYBR qPCR Mix (TOYOBO QKD-201) according to manufacturer instructions.

### 3.11. Neurogenesis in the Hippocampal Dentate Gyrus

C57BL/6J mice were purchased at 15 weeks old and reared under sterile water ad libitum until 26 weeks of age. Based on blood oxidative stress level (d-ROMs test, Diacron International Srl) and blood antioxidant capacity measured by the BAP test and OXY adsorbent test (both Diacron International Srl), mice were divided into the control group (n = 5) and the TwX group (n = 5). The TwX group was fed ad libitum with TwX mixed in drinking water at 40 mg/kg/day.

Additionally, C57BL/6J mice were purchased at 5 weeks old and acclimatized for 1 week to serve as a comparison group of young mice (n = 5). Mice were reared until 56 weeks old and intracardially perfused with PBS. Brains were then perfused with 4% paraformaldehyde in PBS (PFA; pH 7.4–7.5), post-fixed with PFA, and sectioned into 10-µm coronal sections using a microslicer (Dousaka EM, Kyoto, Japan).

The obtained sections were subjected to immunostaining targeting tubulin III (TuJ-1) as previously reported [64]. The number of positive cells was measured by Prof. Tatsuki Ito of Kinki University.

## 4. Conclusions

TwX has shown promising effects by disrupting the persistent cycle of inflammation and mitochondrial dysfunction by suppressing ROS. In this study, we observed that TwX not only reduced elevated blood glucose levels but also prevented telomere shortening and the loss of autophagy function associated with aging and hyperglycemia. Additionally, TwX exhibited protective effects on mitochondria and enhanced energy production by increasing antioxidant capacity. The findings of this study strongly suggest that TwX may have the potential to improve the pathological conditions observed in mouse models of AD and cerebral infarction. Other results also indicated that TwX improved neurovascular dysfunction, reduced phosphorylated tau and a-synuclein levels, and prevented dementia in individuals with MCI. TwX, with its novel blend of antioxidants, can potentially contribute to the prevention of AD in the future.

## Figures and Tables

**Figure 1 ijms-24-13018-f001:**
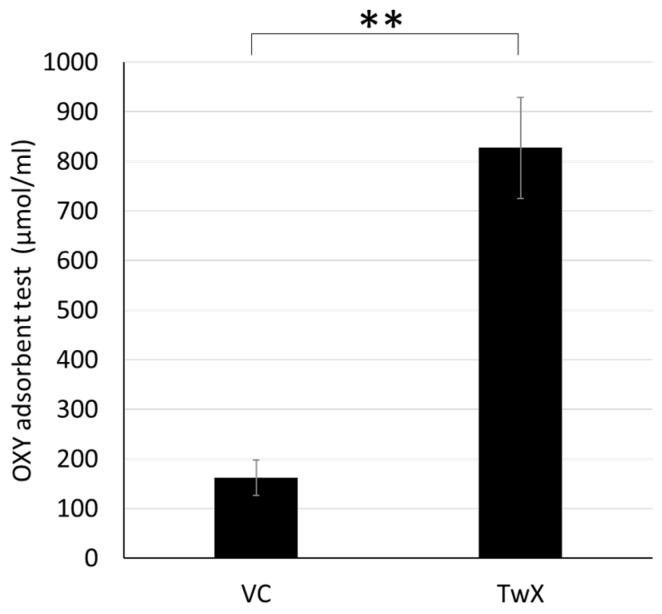
In vitro antioxidant capacity of Twendee X^®^ (TwX) and vitamin C (VC) alone in TwX solution. The antioxidant capacity of TwX (60 mg/mL, n = 6) and VC alone (20.5 mg/mL, n = 6) in the same concentration as is present in TwX was measured by the OXY adsorbent test (Diacron International Srl, Grosseto, Italy). Values in the graph represent the mean ± SD. **: *p* < 0.01 (Student’s *t*-test).

**Figure 2 ijms-24-13018-f002:**
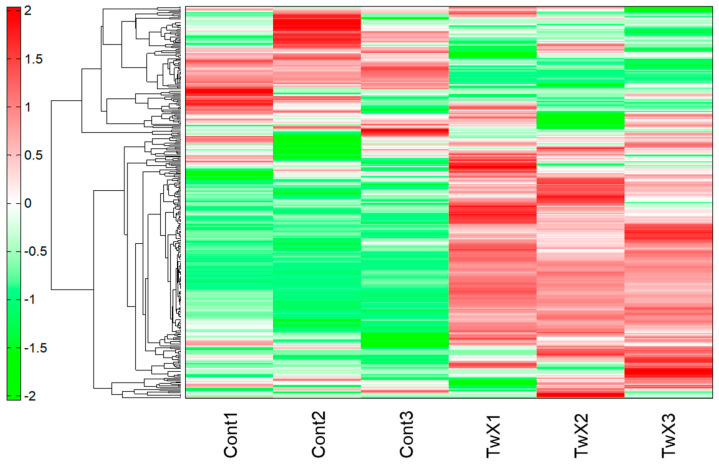
Results of hierarchical cluster analysis (HCA) of HepG2 metabolites. Twendee X^®^ (TwX) at 60 µg/mL was added to the human liver cancer cell line HepG2, and metabolites were sampled after one hour. Unstimulated (Cont1-3) and TwX-stimulated cells (TwX1-3) were compared. The horizontal axis and vertical axis indicate the sample name and the peaks, respectively. HCA was performed on the peaks, and the distance between peaks is shown in the tree diagram in the figure. Darker green or red indicates smaller or larger than average, respectively. The analysis was performed by HMT’s proprietary MATLAB program.

**Figure 3 ijms-24-13018-f003:**
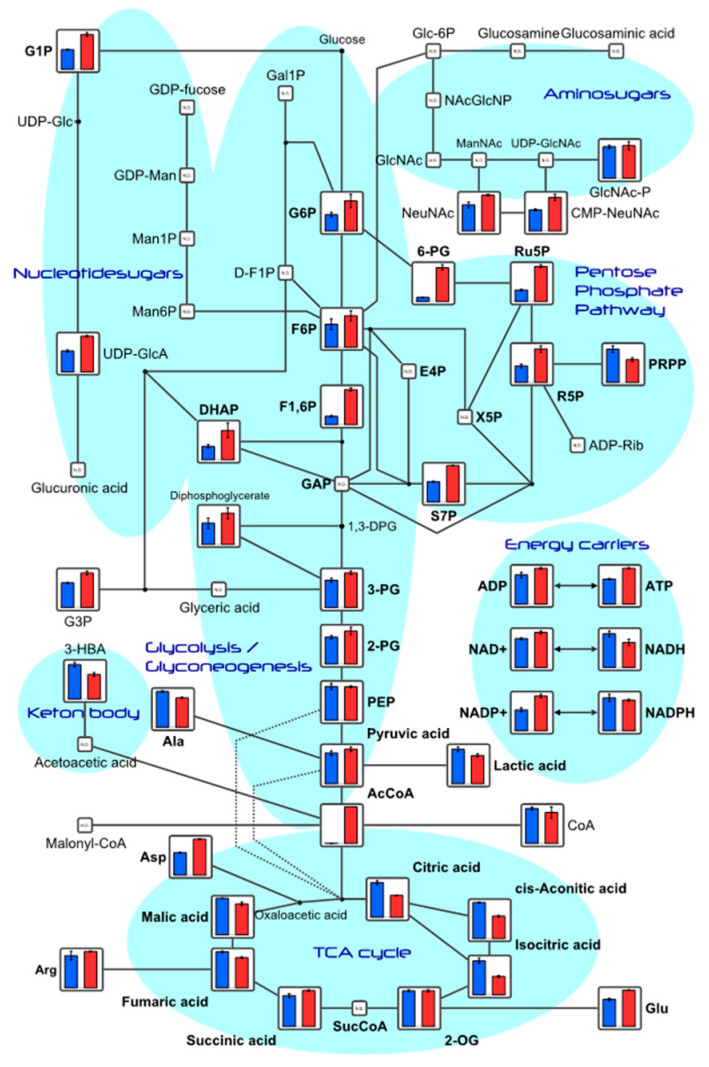
HepG2 metabolite levels near the glycolytic and TCA circuits. Metabolites detected in this study are plotted in the metabolic pathway using VANTED software version 2.6.3. The graph shows relative area values for Control (blue) and Twendee X^®^ (red). Values in the graph represent the mean ± SD. N.D.: not detected. 1,3-DPG, 1,3-Diphospoglycerate; 2-OG, 2-Oxoglutaric acid; 2-PG, 2-Phosphoglyceric acid; 3-HBA, 3-Hydroxybutyric acid; 3-PG, 3-Phosphoglyceric acid; 6-PG, 6-Phosphogluconic acid; AcCoA, Acetyl coenzyme A_divalent; ADP, Adenosine diphosphate; ADP-Rib, ADP-ribose; Ala, Alanine; Arg, Arginine; Asp, Aspartic acid; ATP, Adenosine triphosphate; CMP-NeuNAc, Cytidine-5′-monophosphate--N-acetylneuraminate; CoA, Coenzyme A_divalent; DHAP, Dihydroxyacetone phosphate; E4P, Erythrose 4-phosphate; F1,6P, Fructose 1,6-diphosphate; D-F1P, Fructose 1-phosphate; F6P, Fructose 6-phosphate; Gal1P, Galactose 1-phosphate; GDP-fucose, Guanosine diphosphate-fucose; GDP-Man, Guanosine diphosphate-mannose; Glu, Glucose; Glc-6P, Glucosamine 6-phosphate; G1P, Glucose 1-phosphate; G6P, Glucose 6-phosphate; GAP, Glyceraldehyde 3-phosphate; G3P, Glycerol 3-phosphate; Man1P, Mannose 1-phosphate; Man6P, Mannose 6-phosphate; Malonyl-CoA, Malonyl coenzyme A_divalent; GlcNAc, N-Acetylglucosamine; GlcNAc-P, N-Acetylglucosamine 1-phosphate; NAcGlcNP, N-Acetylglucosamine 6-phosphate; ManNAc, N-Acetylmannosamine; NeuNAc, N-Acetylneuraminic acid; NAD, Nicotinamide adenine dinucleotide; NADH, Reduced nicotinamide adenine dinucleotide; NADP, Nicotinamide adenine dinucleotide phosphate; NADPH, Reduced nicotinamide adenine dinucleotide phosphate_divalent; PEP, Phosphoenolpyruvic acid; PRPP, Phosphoribosyl pyrophosphate; R5P, Ribose 5-phosphate; Ru5P, Ribulose 5-phosphate; S7P, Sedoheptulose 7-phosphate; SucCoA, Succinyl coenzyme A_divalent; UDP-Glc, Uridine diphosphate-glucose; UDP-GlcA, Uridine diphosphate-glucuronic acid; UDP-GlcNAc, Uridine diphosphate-N-acetylglucosamine; X5P, Xylulose 5-phosphate.

**Figure 4 ijms-24-13018-f004:**
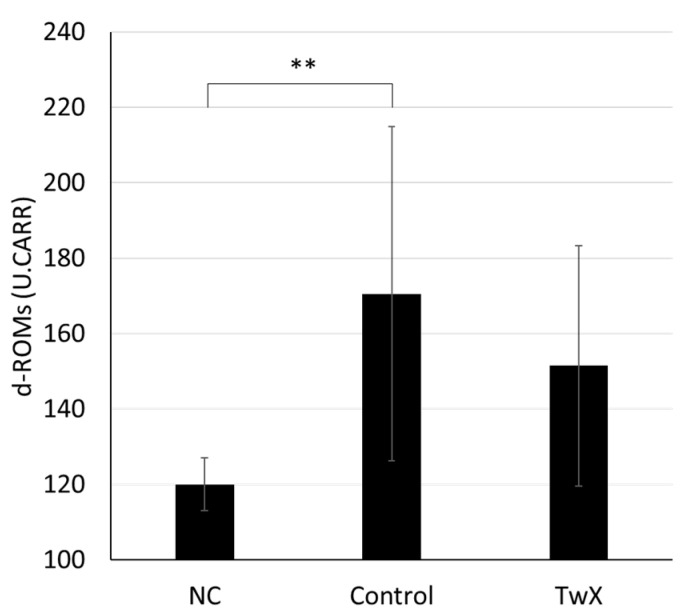
Fasting blood oxidative stress in a mouse model of spontaneous diabetes mellitus. C57BL/6J spontaneous diabetic mice were administered sterile water (Control, n = 10) or Twendee X^®^ (TwX, n = 8) orally at 40 mg/kg/day. Untreated mice of the same age were kept as negative controls (NC, n = 2). After fasting for 16 h, the blood oxidative stress level was measured by the d-ROMs test (Diacron International Srl, Grosseto, Italy). Values in the graph represent the mean ± SD. **: *p* < 0.01 (Student’s *t*-test, NC vs. Control).

**Figure 5 ijms-24-13018-f005:**
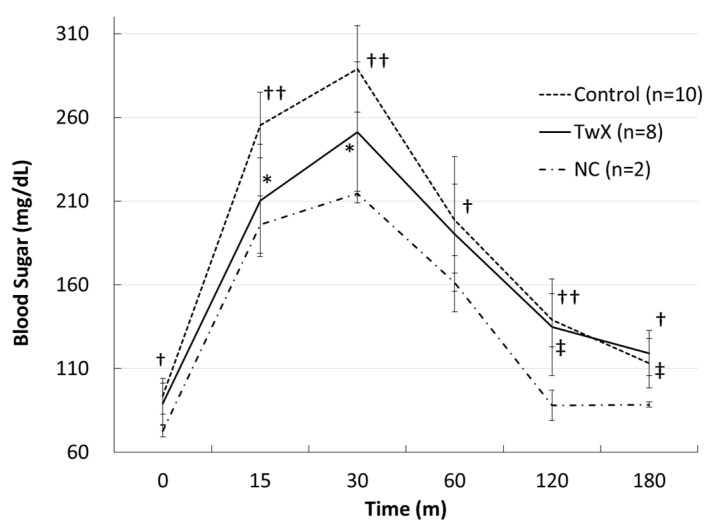
Blood glucose level on the glucose tolerance test in hyperglycemic mice. C57BL/6J spontaneous diabetic mice were administered sterile water (Control) or Twendee X^®^ (TwX) orally at 40 mg/kg/day. Untreated mice of the same age were kept as negative controls (NC). After fasting for 16 h, mice were given glucose, and their blood glucose levels were measured at set intervals. Values in the graph represent the mean ± SD. Student’s *t*-test was used for all statistical analysis. †: *p* < 0.05 (Control vs. NC), ††: *p* < 0.01 (Control vs. NC), ‡: *p* < 0.05 (TwX vs. NC), *: *p* < 0.05 (TwX vs. Control).

**Figure 6 ijms-24-13018-f006:**
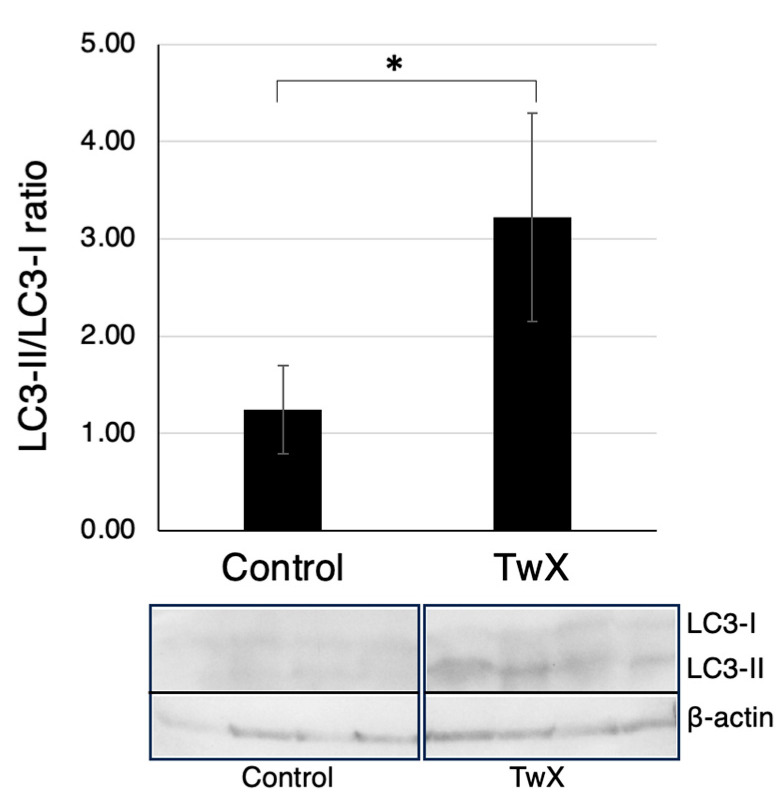
Retention of autophagy in an aged mouse model of spontaneous diabetes mellitus. C57BL/6J spontaneous diabetic mice were administered sterile water (Control) or Twendee X^®^ (TwX) orally at 40 mg/kg/day. At 70 weeks old, total protein from the mouse hippocampus was extracted, and western blot was used to evaluate the protein levels of LC3-I and LC3-II. Control: n = 4, TwX: n = 4. All lanes were loaded with 200 ug of sample. Values in the graph represent the mean ± SD. *: *p* < 0.05 (Student’s *t*-test). Band intensity was evaluated using ImageJ version 1.53k (National Institutes of Health, Bethesda, MD, USA).

**Figure 7 ijms-24-13018-f007:**
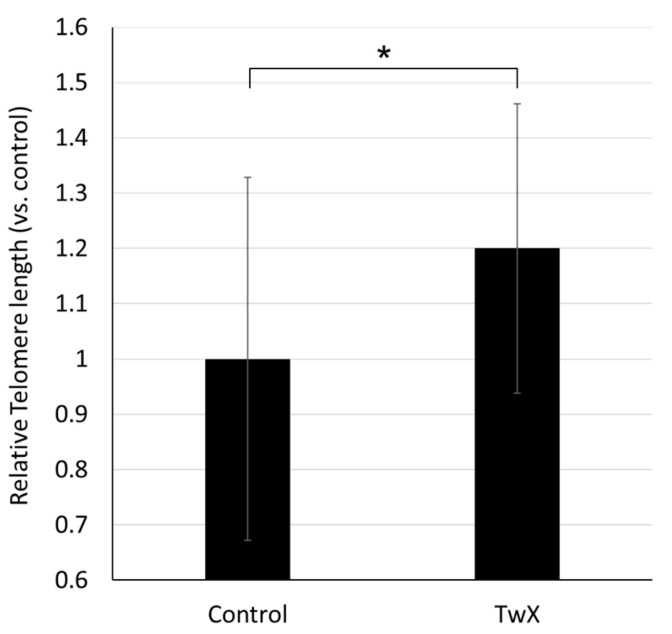
Effect of Twendee X^®^ (TwX) on mouse telomeres. Spontaneous diabetic model C57BL/6J mice were orally administered sterile water (Control, n = 10) or TwX (40 mg/kg/day, n = 10) for 33 weeks, after which telomere length in the tail tissue was measured. Values in the graph represent the mean ± SD. *: *p* = 0.038 (Student’s *t*-test).

**Figure 8 ijms-24-13018-f008:**
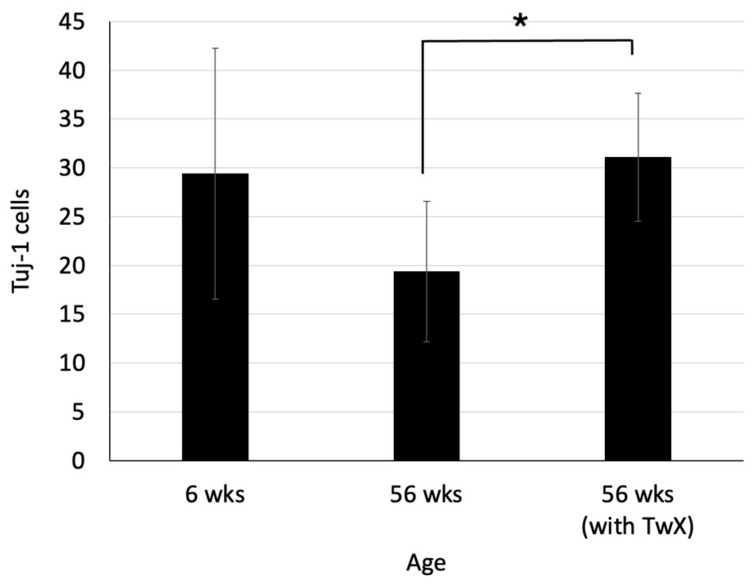
Changes in the number of newborn neurons in the hippocampal dentate gyrus. C57BL/6J mice in the control (sterile water, n = 5) and Twendee X^®^ group (TwX, 40 mg/kg/day, n = 5, administered from 26 weeks old) were dissected at 56 weeks old. Six-week-old C57BL/6J mice were dissected as a control (n = 5). The number of Tuj-1 cells in the hippocampal dentate gyrus was measured. Values in the graph represent the mean ± SD. *: *p* = 0.029 (Student’s *t*-test, 56 weeks vs. 56 weeks (with TwX)).

**Table 1 ijms-24-13018-t001:** Effect on the redox status of HepG2 cells after a 1-h exposure to H_2_O_2_ at 100 µM.

	Dose	Effects on REDOX Status				
		mtROS	cROS	Mn-SOD	Cu/Zn-SOD	GSSG/GSH
H_2_O_2_	100 µM	↑ * 69%	↑ *** 68%	↓ *** 32%	↓ *** 31%	↑ * 31%
TwX	60 µg/mL	↓ 63%	↓ 45%	↑ 147%	↑ 60%	↓ 40%
	120 µg/mL	↓ 77%	↓ 49%	↑ 104%	↑ 33%	NS ↓ 15%
	240 µg/mL	↓ 65%	↓ 37%	NS ↑ 38%	NS ↑ 19%	↓ 20%

H_2_O_2_ 100 µM was administered in HepG2 cells with or without Twendee X^®^ (TwX). Significance levels compared to the non-treated cells. Values are shown as relative absorbance values compared to the non-treated group. ↓/↑: Decrease/increase from baseline. * *p* < 0.05, *** *p* < 0.001, NS: non-significant (Student’s *t*-test). Prefixes: c: cellular, mt: mitochondrial. —% effect is expressed with respect to non-treated cells for the H_2_O_2_ condition and with respect to H_2_O_2_-induced oxidative stress for anti-oxidant test elements. All data were recorded by the SkanIt software version 4.1 (Thermo).

## Data Availability

The data are available upon proper request.

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
