# Peer review of "Why Does the Antioxidant Complex Twendee X® Prevent Dementia?"

_ijms, 2023, doi:10.3390/ijms241613018_

Round 1
Reviewer 1 Report
1. The title is not suitable for the work performed here. The author has given” Why an antioxidant complex Twendee X®ï¸Ž prevent Dementia? The author explains the antioxidant capacities, mitochondrial functions, and effect of Twx on diabetes, autophagy and telomere. But to test dementia, no cognitive or neuropsychological tests were performed in the mouse model after treating TwX. What is the purpose of mentioning dementia in the title? It would be better to change the title related to the work.
2. In lines 147-148 , the author mentioned Twx had a higher antioxidant capacity than a single equal dose of VC. Twx is a mixture. The use of VC as a control is not acceptable. In general, experiments are not having proper controls.
3. Clarify whether the total glutathione level (the sum of GSH + GSSG) or GSSG/GSH ratio has been mentioned in the manuscript. Representing total glutathione (GSSG/GSH) is misleading.
4. In section 2.4 the western blot image is not clear. The author mentioned, Mouse brain lysate loaded as control. Mention here which housekeeping protein used as a control for western blot in section 2.4 and 3.9 as well.
5. In Section, 2.7 Effects on brain functions: Author merely mentioned previous studies as review part. Since author haven’t done any work on the effect of brain functions in this paper, this section can be removed better. If the author really needs to add information, they can use it in the discussion or introduction.
6. Section 4 Conclusion: The conclusion can be improved. The authors did so many tests with TwX, but the conclusion seems very simple, and the author can explain more about the role of TwX they obtained from their results here.
7. A complete pathway figure is required; Figure 3 needs to give concluding information.
8. Dementia is a condition where mental ability is declined, which affect memory and other functions. Oxidative stress may be one of the causes to induce dementia in neurodegenerative diseases. but treating oxidative stress alone could not reduce dementia. Please repharse the line “ TWx with its novel blend of antioxidants can potentially contribute to the prevention and treatment of AD in future”. Treating oxidative stress alone could not a treatment for AD / dementia. Author could improve their conclusion and remove this line.
9. Some minor errors need to be rectified, like long unclear sentences, for example, Line 18-22: Revise the sentences.
10. Below all the graphs, maps, figures the author can mention in which software they prepared them.
Reviewer 2 Report
In this review the authors show mechanisms by which TwX could prevent dementia. The authors employ several experimental strategies. The authors show the antioxidant effect of TwX, which is a supplement containing 8 components, among them vitamin C. This TxW antioxidant effect is much greater than that presented by Vitamin C at the same concentration at which it is presented in TwX. The authors also present a metabolomic study to determine the effect of the supplement on energy production. In vivo, the authors show the effect of the supplement on oxidative stress in the blood of hyperglycemic 35-week-old mice, finding a tendency to decrease oxidative stress levels, but it was also able to significantly decrease blood glucose levels. In 70-week-old mice, the supplement was able to induce LC3 expression while this was not observed in the control. In 65-week-old mice, TwX increased telomere length, as well as increased neurogenesis compared to those not treated with the supplement. However, several points will need to be addressed before this manuscript can be considered for publication:
1. The title is not consistent with what is presented in the manuscript, the authors do not show experimental evidence that the administration of TwX prevents cognitive alterations.
2. Figure 1 where the antioxidant capacity of TwX is determined does not contain a standard deviation or error and considering that it was only n=2 in one of the groups, there is not the number of experiments necessary to carry out a statistical test.
3. Subsequently, they evaluate the redox effect of TwX in HepG2 cells exposed to hydrogen peroxide. The first comment in this regard is why they used this carcinoma cell line, if it is known that cancer cell lines have an accelerated metabolism and increased redox capacity. They should not have used a primary culture of neurons or astrocytes.
4. In table 1, what were the basal levels of ROS, SOD and GSH in the HepG2 line without treatment and against which all the data shown in the table are being compared.
5. GSH levels should be expressed as follows: GSH/GSSG ratio.
6. In Figure 4, the NC+TwX group is missing, and the bars show no standard deviation. Because the authors use a Student's t-test instead of another statistical test if they present 3 experimental groups.
7. In figure 5 it is necessary to add the deviation in each of the times, in addition to adding the group NC+TwX
8. It is not clear the number of samples that were used for the determination of LC3, why isn't there a graph derived from this expression?
9. Figure 7 It is necessary to add standard deviation.
10. Figure 8, the 6ws+TwX group is missing, as well as the standard deviations and a correct statistical analysis.
11. Describe in detail how SOD activity and GSH levels are assessed.
Reviewer 3 Report
The authors investigated about the effects of Twendee X®ï¸Ž on accelerated cognitive decline, and also on diabetes, brain autophagy, neurogenesis and telomere length.
The rational behind the study was clear and straight forward. The manuscript is almost well written. Overall the topic could be interesting but some details could be improved.
I recommend that the paper be accepted with minor revision:
a) The authors should mentioned in the abstract more details about model used.
b) In the introduction section, little previous evidence is provided about the importance of AD in daily life. Incorporating comparisons with other studies would increase the strength of the paper. Please refer to doi: 10.3390/antiox10111664; 10.1038/s41422-021-00582-x; 10.3390/antiox10050818; 10.4103/1673-5374.239448; 10.3233/JAD-200675.
c) In the introduction section, the authors should explain the connection between AD and diabetes.
d) The authors should clarify why they use this dose of TwX (40mg/kg/day). Any reference?
e) How the animals were euthanized? Clarify
f) There are some minor grammar issues that should be fixed in order to aid the accessibility of the results to the reader.
Minor editing of English language required
Round 2
Reviewer 1 Report
the manuscript is improved compared to the original submission.
Author Response
Thank you for the comment. These reviews were very insightful for us.
Reviewer 2 Report
The authors considered the comments, and the presentation of the results improved the manuscript. Only a few brief points need to be addressed before this manuscript can be considered for publication.
1. I understand that the data shown in table 1 are relative values. However, it is important to mention in the text the real values of the controls so that the reader knows what these values that are found in Table 1 refer to. Therefore, I ask the authors to add this information.
2. In all figure captions it is necessary for the authors to add what they represent in the graphs, that is, the mean +/- standard deviation or standard error.
